# Recent Advances in Encapsulation, Protection, and Oral Delivery of Bioactive Proteins and Peptides using Colloidal Systems

**DOI:** 10.3390/molecules25051161

**Published:** 2020-03-05

**Authors:** Sarah L. Perry, David Julian McClements

**Affiliations:** 1Department of Chemical Engineering, University of Massachusetts Amherst, Amherst, MA 01003, USA; perrys@engin.umass.edu; 2Department of Food Science, University of Massachusetts Amherst, Amherst, MA 01003, USA; 3Department of Food Science & Bioengineering, Zhejiang Gongshang University, 18 Xuezheng Street, Hangzhou 310018, China

**Keywords:** encapsulation, insulin, lipase, lactase, nanoparticles

## Abstract

There are many areas in medicine and industry where it would be advantageous to orally deliver bioactive proteins and peptides (BPPs), including ACE inhibitors, antimicrobials, antioxidants, hormones, enzymes, and vaccines. A major challenge in this area is that many BPPs degrade during storage of the product or during passage through the human gut, thereby losing their activity. Moreover, many BPPs have undesirable taste profiles (such as bitterness or astringency), which makes them unpleasant to consume. These challenges can often be overcome by encapsulating them within colloidal particles that protect them from any adverse conditions in their environment, but then release them at the desired site-of-action, which may be inside the gut or body. This article begins with a discussion of BPP characteristics and the hurdles involved in their delivery. It then highlights the characteristics of colloidal particles that can be manipulated to create effective BPP-delivery systems, including particle composition, size, and interfacial properties. The factors impacting the functional performance of colloidal delivery systems are then highlighted, including their loading capacity, encapsulation efficiency, protective properties, retention/release properties, and stability. Different kinds of colloidal delivery systems suitable for encapsulation of BPPs are then reviewed, such as microemulsions, emulsions, solid lipid particles, liposomes, and microgels. Finally, some examples of the use of colloidal delivery systems for delivery of specific BPPs are given, including hormones, enzymes, vaccines, antimicrobials, and ACE inhibitors. An emphasis is on the development of food-grade colloidal delivery systems, which could be used in functional or medical food applications. The knowledge presented should facilitate the design of more effective vehicles for the oral delivery of bioactive proteins and peptides.

## 1. Introduction

Bioactive proteins and peptides (BPPs) exhibit a diverse range of biological activities and functional attributes that make them suitable as therapeutic agents, such as anti-hypertension, antimicrobial, antioxidant, enzyme, hormone, and immunological activities [1,2,3,4,5,6,7]. Consequently, there is considerable interest in incorporating them into supplements, pharmaceuticals, and functional foods specifically designed to prevent or treat certain chronic conditions. For practical and consumer compliance reasons, it is usually advantageous to administer BPPs via the oral route. There are, however, numerous challenges that must be overcome before these bioactive substances can be orally delivered.

Two main challenges associated with the creation of an orally administrable BPP are the need to maintain the activity of the cargo molecule: (i) during formulation and (ii) during delivery to the site of action. While these two challenges are not specific to oral delivery, they are complicated by the harsh chemical environments present within the human gastrointestinal tract (GIT), such as the strongly acidic and enzymatically active gastric fluids within the human stomach [8]. For these reasons, BPPs may need to be administered using some kind of colloidal delivery system (CDS) to protect them from degradation during storage and passage through the GIT, but then release them at the required location inside the human body [3,9,10,11]. Many kinds of CDS have been created that may be suitable for this purpose, including microemulsion droplets, emulsion droplets, solid fat particles, liposomes, and microgels (Figure 1). Each of these systems has its own benefits and limitations for specific applications. Consequently, it is important to understand the factors impacting their formation, properties, and performance in order to select the most appropriate one for a particular application [12].

This review article provides an overview of the key parameters influencing the rational design of colloidal particles for the oral delivery of BPPs. In particular, an emphasis is given to the development of food-grade colloidal particles that could be used in functional or medical food applications. It should be noted that there are important differences in the nature of the ingredients that can be used to assemble colloidal delivery systems for food and pharmaceutical applications [13]. In the pharmaceutical industry, a broad range of synthetic polymers, lipids, and surfactants are available to assemble colloidal particles with specific functionalities. In the food industry, however, the ingredients available are much more limited. For instance, in the USA, only ingredients that are generally recognized as safe (GRAS) can be utilized. Moreover, many consumers are demanding “clean-label” food products, so manufacturers are restricted to using more natural ingredients rather than synthetic ones, such as proteins, polysaccharides, phospholipids and lipids from plants, meat, eggs, or milk. This makes it much more challenging to create food-based colloidal delivery systems with the required functional attributes.

## 2. Protein Characteristics

The selection of a suitable CDS for a particular application depends on the molecular and physicochemical attributes of the bioactive proteins and peptides to be delivered [12].

### 2.1. Molecular Dimensions

The size and structure of proteins and peptides influences their retention and release within CDSs. The molar mass of BPPs can vary considerably, from around 1000 Da for small peptides to over 100,000 Da for large proteins. For BPPs with the same molar mass, the molecular dimensions in aqueous solution are strongly influenced by their three-dimensional structure: globular proteins < random coil < rigid rod [14]. Furthermore, BPPs may be present as single molecules, small clusters of molecules, or complex hierarchical structures depending on their biological function, the extraction methods used to isolate them, and the processing operations used to treat them [15,16]. As a result, the molecular size of BPPs may vary from a few nanometers to a few hundred nanometers or more. Information about the size of BPPs is critical for identifying and selecting an appropriate CDS. In the case of phase separated CDSs, such as water-in-oil microemulsions, nanoemulsions, or emulsions (Figure 1), the BPPs should have molecular dimensions that are smaller than the dimensions of the water domains if they are going to be successfully encapsulated [17]. On the other hand, in the case of biopolymer microgels (Figure 1), the BPPs should have molecular dimensions that are larger than the dimensions of the pores in the biopolymer network that makes up the interior of this kind of colloidal particle in order to achieve physical entrapment and limit release via diffusion.

### 2.2. Electrostatic Effects

The electrostatic properties of BPPs also influence their functional performance within CDSs, because their retention and release depend on the nature of any electrostatic interactions between the polypeptides and the colloidal particles [18,19,20]. As most BPPs have both anionic and cationic groups, their net charge changes from negative to positive as the pH is reduced from above to below their isoelectric point (pI). The electrical properties of BPPs in aqueous solutions are typically measured using electrophoresis instruments and are conveniently represented by their ζ-potential *versus* pH profile (Figure 2).

Information about the electrical attributes of BPPs is often essential for the design of an efficacious CDS. As an example, the retention/release of BPPs from biopolymer microgels is strongly influenced by the electrical interactions between the proteins and the biopolymer network inside the microgels. BPPs are electrostatically attracted to anionic biopolymers, like alginate, carrageenan, or pectin, when the pH is less than their isoelectric point, but they are electrostatically repelled when the pH is above their isoelectric point [21,22]. As a result, they may be retained at low pH values, but released under high pH values due to the change in electrostatic interactions. The opposite phenomenon occurs for cationic biopolymers, such as chitosan or polylysine. The magnitude of any electrostatic interactions in aqueous solutions is reduced when dissociable salts are added as a result of electrostatic screening, i.e., accumulation of salt counter-ions around charged groups on the proteins [23]. This has important practical implications because it means that it may be challenging to keep BPPs trapped within the interior of biopolymer hydrogels using electrostatic attraction in commercial products that contain salts. Conversely, it means that it may be possible to develop CDSs that can release proteins in response to changes in the ionic strength of their environment.

Beyond net charge considerations, it is important to note that the complex chemical and physical nature of many BPPs means that the spatial arrangement of the charges can also be important in dictating their interactions with CDSs [24,25,26]. For example, serum proteins such as bovine serum albumin (BSA) tend to have a uniform charge distribution, while lysozyme has a cluster of cationic residues on its surface. This clustering of cationic charge has been shown to drive nearly 100-fold higher loading of lysozyme into microgels formed from equimolar mixtures of oppositely-charged polymers than for BSA [27].

### 2.3. Polarity, Solubility, and Surface Activity

The polarity of BPPs is another critical factor influencing their ability to be encapsulated, since it impacts their three-dimensional structure, solubility, surface activity, and molecular interactions. BPPs may be predominantly polar, non-polar, or amphiphilic depending on the number and distribution of hydrophilic and hydrophobic amino acids in the polypeptide chain, which in turn influences their structural arrangement in aqueous solutions. Polar groups are able to form dipole-dipole interactions with water, whereas non-polar ones are not. A major driving force for protein folding is the tendency to reduce the number of hydrophobic non-polar groups exposed to water [28]. As a result, BPPs may be either soluble or insoluble in aqueous solutions depending on their surface polarities. The surface activity of BPPs depends on the distribution of polar and non-polar groups on their surfaces. Many polypeptides are amphiphilic molecules that are able to adsorb to air-water, oil-water, or solid-water interfaces, which allows them to be utilized as functional ingredients to stabilize foams, emulsions, or suspensions [29].

### 2.4. Stability

The physical and chemical stability of BPPs is important because it impacts their functionality [30,31]. The three-dimensional structure and functionality of proteins may be irreversibly altered by environmental factors, such as changes in pH, ionic composition, solvent quality, temperature, pressure, or adsorption to surfaces. It is therefore important to identify and specify the major factors impacting the stability of the BPPs one is trying to encapsulate, such as the temperatures or pH values where they become denatured. In many cases, CDSs are specifically designed to enhance the stability of BPPs by encapsulating them within protective environments.

## 3. Hurdles to the Oral Delivery of Proteins

Various challenges have to be overcome when designing oral delivery systems for BPPs [32,33].

### 3.1. Delivery Vehicle Compatibility

BPPs may be encapsulated within functional foods, supplements, medical foods, or pharmaceutical preparations that differ in their physicochemical attributes and storage conditions. As an example, the delivery vehicle may be a fluid, a gel, a powder, a capsule, or a tablet. Moreover, these products can experience a range of temperatures, light exposures, oxygen levels, and humidity throughout their lifetime. The delivery vehicle must therefore be carefully designed to ensure that the BPPs are effectively encapsulated without negatively impacting the desirable quality attributes (appearance, texture, and taste) of the product, as well as remaining stable during production, transportation, storage, and application [34,35,36,37].

### 3.2. Stability in Gastrointestinal Tract

After ingestion, the three-dimensional structure and functionality of BPPs may be altered as they move through the human gut. For instance, they may undergo hydrolysis, structural rearrangements, or aggregation when exposed to the fluids within the GIT [35,37]. The gastrointestinal fluids vary greatly in pH throughout the GIT, ranging from highly acidic in the stomach to neutral or slightly basic in the duodenum [38,39]. There are also various kinds of digestive enzymes (proteases) that can hydrolyze BPPs and alter their functionality [40,41]. Finally, the gastrointestinal fluids contain biological surfactants (bile salts and phospholipids) that may bind to BPPs and alter their biological activity. The stability of BPPs within the GIT can often be improved by trapping them inside colloidal particles, thereby isolating them from the stressors in the gastrointestinal fluids. This requires that the particles be designed so that they do not breakdown until they reach the targeted region (such as mouth, stomach, small intestine, or colon, depending on the application). Moreover, the particles may have to be designed to be impermeable to stressors in the gastrointestinal fluids (such as bile salts or digestive enzymes). Otherwise, these substances may penetrate into the particles and degrade the encapsulated BPPs. On the other hand, changes in the integrity or permeability of the particles can be used to deliver BPPs to different regions of the GIT. For instance, colloidal particles that do not degrade in the upper GIT but do degrade in the large intestine (such as those made from dietary fibers or certain synthetic polymers) can be used to deliver BPPs to the colon.

### 3.3. Absorption from Gastrointestinal Tract

The biological activity of BPPs may also be limited because they are not efficiently absorbed by the body [35,37]. BPPs have to be released from any matrix they are trapped in, then travel through the gastrointestinal fluids and mucus layer before being absorbed by the epithelium cells (Figure 3). Their transport rate and residence time will depend on the viscosity of the gastrointestinal fluids surrounding them, which can be modulated by adding components such as thickening agents [42,43,44]. The BPPs, or the colloidal particles containing them, must be small enough to penetrate through the mucus layer [45,46,47]. The particles in some CDSs are relatively large (>500 nm) and are therefore too large to penetrate through the mucus layer intact. Nevertheless, they may be dissociated or degraded within the GIT, thereby releasing the BPPs, which can then diffuse through the mucus layer. Once they reach the surface of the epithelium cells the BPPs may be absorbed through numerous mechanisms, such as active or passive transcellular routes, paracellular (tight-junctions) routes, or endocytosis [11,33]. Moreover, the BPPs may be absorbed by different kinds of epithelium cells (such as enterocytes or M-cells) depending on their size and surface chemistry. M-cells, which are located in the Peyer’s patches, are typically much more effective at absorbing colloidal particles (*d* < 500 nm) than enterocytes, but they are much less prevalent in the GIT, which limits their effectiveness for this purpose [11]. Many kinds of proteins and BPP-loaded colloidal particles are too large or too hydrophilic to be absorbed by the mechanisms typically used for smaller pharmaceuticals or nutraceuticals. For instance, they are too large to pass through the tight junctions and too hydrophilic to pass through the phospholipid bilayers that make up the epithelium cell membranes. Moreover, efflux inhibitors are typically less effective for BPPs than for small hydrophobic molecules (with the exception of the peptide cyclosporine).

Because the absorption of many BPPs is inherently low, special strategies have to be developed to increase it e.g., permeation enhancers, mucoadhesive materials, or colloidal carriers can be utilized [11,48]. Permeation enhancers can increase the absorption of substances in the GIT by temporarily disrupting the intestinal barrier, by increasing membrane fluidity, or by opening the connections (tight junctions) separating the epithelium cells [11,49]. For instance, the pharmaceutical company Chiasma has developed a Transient Permeability Enhancer (TPE^®^) technology to increase the absorption of macromolecules by opening up the tight junctions. An important consideration in the design of these permeation enhancers is to ensure that they allow the BPPs through, but do not promote the absorption of undesirable toxins or microbial pathogens. Other types of permeation enhancer are also used commercially to increase the absorption of BPPs within the human gut. For example, semaglutide (Rybelsus^®^, Novo-Nordisk, Plainsboro Township, NJ, USA) is an orally administered drug designed to treat Type 2 diabetes, which contains a permeation enhancer (salcaprozate sodium) to increase the amount of a GLP-1 receptor agonist absorbed by the body [48,50,51]. Some CDSs naturally contain components (such as surfactants or medium chain fatty acids) that can increase the permeability of cell membranes, which may also be useful for increasing BPP absorption [50].

Typically, it is difficult to create colloidal particles that are able to retain the BPPs as they travel through the GIT tract and then be absorbed intact by the epithelium cells. This is because only a very small fraction of the colloidal particles is typically absorbed by the body. The fraction of colloidal particles that is absorbed depends on their size and charge [52,53]. Hence, it may be possible to design more effective BPP-loaded colloidal particles by carefully controlling these parameters. Incorporating protease inhibitors in CDS can help to reduce the hydrolysis of BPPs in the gut (but can also have adverse effects on normal digestion processes), whereas coating colloidal particles with mucoadhesive materials can increase their retention in the small intestine, thereby allowing more time for absorption to occur [11].

### 3.4. Product Requirements

After the properties of the BPPs have been specified, and the challenges limiting their potential efficacy have been established, then the requirements of the end product that will be used to deliver them needs to be defined, such as a functional food, medical food, supplement, or pharmaceutical preparation. Attributes such as the look, feel, taste, and shelf life of the end product should be determined and specified [54]. Obviously, different end products have different required product attributes. For instance, a functional food may be in the form of a fluid beverage or a soft cereal bar, whereas a supplement or drug may be in the form of a hard pill or soft capsule. When deciding which end product is most suitable for a particular BPP, it is important to consider its optical properties (which may go from clear to opaque), its rheological properties (which may go from fluid to solid), its behavior in the mouth (such as flavor profile, disruption/dissolution, and residence time), and the environmental stresses it experiences during its lifetime (such as heat, moisture, light, oxygen, and mechanical stresses) The BPPs should be compatible with the end product matrix, they should remain stable throughout the lifetime of the product, they should be present at a sufficiently high dose, and they should be stable within the mouth and gastrointestinal tract until they reach the required absorption site.

## 4. Characteristics of Colloidal Particles

The characteristics of the colloidal particles used to encapsulate BPPs will depend on their specific properties, as well as the nature of the end product used to administer them [54]. The most important characteristics of colloidal particles influencing their ability to encapsulate, protect, and deliver BPPs are briefly outlined here.

### 4.1. Composition

The colloidal particles used to encapsulate BPPs can be produced using a range of different edible ingredients, such as proteins, polysaccharides, lipids, phospholipids, and surfactants [55,56,57]. These ingredients influence the functional attributes of the colloidal particles (such as their ability to encapsulate, protect, retain, and release the BPPs). Consequently, the selection of the most appropriate ingredients to fabricate a CDS is a crucial decision. As an example, colloidal particle composition determines the region in the GIT where they are digested and release the BPPs. For instance, starches may be digested within the mouth (by amylases), proteins and lipids within the stomach and small intestine (by proteases and lipases), and dietary fibers in the colon (by microbial enzymes). The nature of the ingredients used may also be impacted by labeling or regulation issues. For some products, it is important to select ingredients that are suitable for particular populations, such as those with Kosher, non-GMO, vegan, vegetarian, or non-allergenic dietary needs. The economics, stability, ease of use, and consistency of the ingredients used may also have to be accounted for.

### 4.2. Particle Morphology and Dimensions

The morphology and dimensions of the colloidal particles utilized to encapsulate BPPs must also be controlled for specific applications. It is possible to create colloidal particles that range in diameter from about 10 nm to 1 mm depending on the nature of the ingredients and manufacturing processes utilized to assemble them. In most cases, colloidal particles are spherical, but other morphologies are also possible, including ellipsoid, cubical, fibrous, or irregular. The size and shape of the particles in a CDS influence their optical, rheological, stability, retention, release, and absorption characteristics. For instance, colloidal dispersions containing particles with dimensions < 50 nm tend to be optically clear and have very good stability to aggregation and gravitational separation. Conversely, colloidal dispersions containing particles with dimensions from around 1 to 10 μm tend to be turbid, and be highly prone to gravitational separation. The retention/release characteristics of CDSs can be controlled by manipulating the particle dimensions, with greater retention and slower release for larger particle sizes. The degradation rate of enzyme-digestible particles (such as those made from lipids, proteins, or starches), tends to increase as their particle size decreases because this increases their specific surface area. Finally, the penetration through the mucus layer and the absorption by the epithelium cells also depends on particle dimensions, with smaller particles typically having a higher permeability.

### 4.3. Interfacial Characteristics

The interfacial characteristics of colloidal particles, such as their chemistry, polarity, charge, rheology, and thickness, play an important role in many of their functional attributes, such as their physicochemical stability and interactions with surfaces. These characteristics can be controlled by assembling the colloidal particles from different ingredients. The surface charge can be manipulated by adsorbing charged emulsifiers or biopolymers to their surfaces, which can be utilized to tailor their functionality for particular applications. For instance, the surface charge can be controlled to modify their mucoadhesive properties: positively-charged particles bind strongly to the negatively-charged mucus layer that lines the GIT, thereby giving them more time to release the BPPs.

### 4.4. State of Aggregation

Colloidal particles may be present as individual entities that are evenly spread throughout a system, or they may be present as clusters that vary in their size and shape. The state of aggregation of colloidal particles often influences their functional performance. For instance, the formation of clusters may decrease the stability to gravitational separation, increase the viscosity, reduce the gastrointestinal digestibility of colloidal particles, or reduce their ability to penetrate through the mucus layer [58,59]. For these reasons, it is often critical to carefully control the state of aggregation of colloidal particles, which usually involves manipulating the colloidal interactions between them (such as electrostatic or steric repulsion).

## 5. Functional Performance of Colloidal Particles

The functional performance of colloidal delivery systems can be defined in terms of their ability to encapsulate, retain, protect, and release BPPs [12].

### 5.1. Loading Capacity and Encapsulation Efficiency

The loading capacity (LC) and encapsulation efficiency (EE) are two of the most important characteristics of CDSs [60]. The LC is a measure of the maximum amount of the BPPs that can be loaded into a particular delivery system, whereas the EE is a measure of the fraction of the BPPs in the system that are actually trapped inside the colloidal particles:LC = *m*_B,E_/*m*_P_(1)
EE = 100 × *m*_B,E_/*m*_B,T_(2)

Here, *m*_B,E_ is the mass of the BPPs encapsulated inside the colloidal particles, *c*_P_ is the mass of the colloidal particles (BPPs + carrier material), and *m*_B,T_ is the total mass of the BPPs in the system (encapsulated and non-encapsulated). The LC is often expressed as μg of encapsulated BPPs per mg of colloidal particles, whereas the EE is expressed as the percentage of the added BPPs trapped inside the particles. The values of LC and EE for a particular colloidal system are governed by the molecular and physicochemical attributes of the BPPs (particularly the oil-water partition coefficient), in addition to the nature of the carrier material used (particularly the permeability). Many BPPs are predominantly polar and so colloidal particles need to have some hydrophilic regions inside them, such as those found in reverse micelles, W/O microemulsions, W/O emulsions, W/O/W emulsions, liposomes, and microgels (Figure 1). Some BPPs are predominantly hydrophobic, or at least have appreciable hydrophobic patches on their surfaces, and so they may be held inside colloidal particles throughout hydrophobic attraction. Many BPPs are electrically charged and can therefore be trapped inside colloidal particles by binding to oppositely charged groups. The sign and magnitude of the electrostatic interactions between BPPs and carrier materials can often be manipulated by altering solution conditions (such as pH or salt concentration), which may be advantageous for the development of triggered release systems.

### 5.2. Retention/Release

In some applications, it is desirable to retain the BPPs inside the colloidal particles until they are exposed to specific conditions that trigger their release, e.g., a variation in solution pH, salt concentration, enzyme activity, or temperature [60]. The BPPs may be released from the colloidal particles via various mechanisms, including simple diffusion, swelling, surface erosion, dissociation, or changes in molecular interactions (Figure 4). It should be noted that once the BPPs are released, they may no longer be protected from any stressors within their environment (such as acids or proteases). For this reason, the colloidal particles must be carefully designed to retain the BPPs until they reach an environment where they can survive long enough to exhibit their beneficial biological effects, which may be inside the gut or the body.

#### 5.2.1. Simple Diffusion

BPPs may simply be released from colloidal particles as a result of diffusion–the BPPs move from inside the particles to outside due to the concentration gradient. In this case, the rate of release increases as the diffusion coefficient of the BPPs through the carrier matrix increases, and as the dimensions of the colloidal particles decrease [61]. For polymeric colloidal particles, the diffusion coefficient can be controlled by manipulating the pore size of the polymer network–the smaller the pore size, the slower the release [62,63]. Typically, the pore size must be smaller than the size of the BPPs to greatly reduce the release rate, which is often difficult to achieve for individual proteins and peptides because they are so small (<5 nm).

#### 5.2.2. Swelling

In some cases, the dimensions of the pores within polymeric colloidal particles can be tuned by altering solution or environmental conditions (like pH, salt content, or temperature) to promote swelling or shrinkage [64,65]. As an example, the pores in colloidal particles fabricated from polyelectrolytes swell when there is a strong electrostatic repulsion between them, but shrink when there is a weak electrostatic repulsion or an electrostatic attraction. The size of the pores can therefore by tuned by increasing or decreasing the ionic strength, or by changing the pH to alter the sign and magnitude of the charge on the polyelectrolytes [66,67]. As a result, it may be possible to develop salt-triggered colloidal delivery systems for BPPs. Polymeric colloidal particles may also swell or shrink in response to an alteration in their temperature, since this may promote a structural change in the polymer chains [65]. For instance, the pores in native starch granules suspended in water are relatively small at ambient temperature, but they increase appreciably when the temperature is raised [68,69]. This type of particle could be used to create temperature-triggered colloidal delivery systems.

#### 5.2.3. Molecular Interactions

The retention and release of BPPs from colloidal particles may also be controlled by manipulating the molecular interactions between them and the carrier material [20,23]. The BPPs will be retained when there is a sufficiently strong attractive force, but released when there is a repulsive force or only a weak attractive force. A frequently employed approach based on this phenomenon is to manipulate the molecular interactions by altering the pH or ionic strength of the surrounding solution to alter the electrostatic interactions [19]. The electrical potential on many proteins goes from negative to positive as the pH is decreased from above to below their isoelectric points (Figure 2). In the case of colloidal particles assembled using anionic polymers (like alginate), the proteins are attracted (retained) at low pH but repelled (released) at high pH [22]. The magnitude of any electrostatic interactions is greatly reduced in the presence of salts because of electrostatic screening [23]. As a result, the BPPs may be released from a colloidal particle if the salt concentration is sufficiently high, even when the proteins and carrier matrix have opposite charges. 

#### 5.2.4. Particle Erosion or Dissociation

Encapsulated BPPs can be released from colloidal particles by designing them to degrade or dissociate when they encounter specific environmental conditions, such as pH, ionic strength, temperature, or enzyme activity (Figure 4) [21]. The environmental responsiveness of colloidal particles can be tuned by assembling them from different types of ingredients. Starches are degraded in the mouth by amylases, proteins in the stomach by proteases, lipids in the small intestine by lipases, and dietary fibers in the large intestine by colonic bacteria [21]. Consequently, the release of encapsulated BPPs may be triggered by a change in enzyme activity when the colloidal particles reach a certain region of the human gut by assembling them from specific ingredients. Colloidal particles can also be designed to fall apart in response to an alteration in the pH or ionic composition of their surroundings, since this may weaken any electrostatic attractive forces holding the molecules inside the particles together [70]. Studies have shown that microgels fabricated from proteins and anionic polysaccharides remain intact when the pH is below the pI value of the protein because of the electrostatic attraction between the cationic protein and anionic polysaccharide [71]. Conversely, the microgels fall apart when the pH is above the pI value because both the protein and polysaccharide are negatively charged and therefore repel each other. However, it may be necessary for the pH to be increased significantly above the pI in order to overcome electrostatic interactions from clusters of charge on the protein [27]. It is also possible to design temperature-triggered colloidal particles that fall apart and release their contents when the temperature is increased above or decreased below a specific value. As an example, studies have shown that microgels fabricated from gelatin and pectin remain intact at room temperature, but fall apart when they are heated, which was linked to a helix-to-coil transition of the gelatin [72,73,74].

One application where control of the retention and release of BPPs is particularly important is for the delivery of proteins or peptides to the colon. The BPPs may be used to target diseases of the large intestine, such as inflammatory bowel or irritable bowel syndromes [75]. Alternatively, BPPs may be used to alter the gut microbiome by selectively promoting the growth of some bacteria and suppressing the growth of others. Colloidal particles can be created that will stay intact within the upper GIT but be degraded by the enzymes released by the bacteria residing inside the colon. Food-grade nanoparticles or microgels can be assembled from dietary fibers, whereas pharmaceutical-grade ones can also be assembled from various kinds of synthetic polymers [75]. A number of BPP-loaded colloidal particles for this purpose are undergoing clinical development, which have recently been reviewed in detail elsewhere [75].

### 5.3. Protection

The stability of BPPs to physical transformations or chemical degradation can sometimes be enhanced by controlling the microenvironment within a colloidal particle. Some of the materials used to construct colloidal particles, including proteins, polyphenols, and chelating agents, exhibit antioxidant activity and so are able to protect encapsulated BPPs from oxidation [76,77]. Other construction materials, including antacids and buffers, are capable of inhibiting pH changes inside colloidal particles, and so are able to enhance the pH-resistance of encapsulated BPPs [78,79]. Some construction materials, including sugars, polyols, salts, and surfactants, can stabilize the native structure of BPPs, and so are able to improve their resistance to denaturation [31,80].

### 5.4. Particle Stability

The particles in CDSs may have to remain stable over a wide range of environmental conditions, including pH changes, salt levels, enzyme activities, light, oxygen, and temperatures. For this reason, they should be carefully constructed so they are stable under all of the environmental conditions they encounter throughout their lifetimes. This means that they have to be designed to resist gravitational separation (usually sedimentation) and aggregation. Gravitational separation can be retarded by reducing the size of the colloidal particles, matching their density to the surrounding liquids, or increasing the viscosity of the surrounding liquids. Aggregation can be prevented by ensuring there is a strong electrostatic or steric repulsion between the colloidal particles.

### 5.5. Particle Permeability

The movement of BPPs throughout the interior of colloidal particles influences their retention and release in a delivery system. In addition, the movement of substances from the surrounding fluids into the colloidal particles, such as ions, enzymes, or surface-active molecules, can negatively impact the stability of the encapsulated BPPs. Molecular movement inside colloidal particles is influenced by various factors, including the rheology, pore size, and interactions of the particle interior [81,82]. Mass transport processes can therefore be controlled by altering the composition and structure of the particle interior. This phenomenon is particularly important for colloidal particles that are comprised of polymer networks, such as microgels.

### 5.6. Potency and Half-Life

Other factors that need to be considered are the potency and half-life of the BPPs encapsulated within the colloidal particles [83]. The potency of a bioactive agent is a measure of its biological activity, which is related to the concentration required to give an effect of specific intensity: the higher the potency, the lower the concentration needed. Commonly, the potency is taken to be the concentration of a bioactive substance required to give the half the maximal effect. It is therefore important that the amount of BPPs reaching the intended site-of-action is high enough to ensure that they are effective, which will depend on the initial dose of the BPPs ingested, as well as any degradation that occurs within the GIT and body. The half-life of the BPPs is related to the time that they remain in the systemic circulation (plasma): it is normally taken as the time for the concentration of the BPPs to decrease by 50%. This value will depend on the properties of the BPPs, as well as any colloidal particles they are encapsulated within.

## 6. Delivery System Selection

The past decade or so has led to a large increase in the development and testing of CDSs for the encapsulation of functional ingredients [12,57,60,84]. In this section, the ones that are most suitable for application with BPPs are briefly outlined. Since most BPPs are predominantly hydrophilic, the focus will be on delivery systems containing colloidal particles with internal hydrophilic domains.

### 6.1. Microemulsions and Emulsified Microemulsions

Microemulsions are a type of colloidal dispersion that is thermodynamically stable [85,86]. They contain small surfactant-based colloidal particles (typically 5 to 100 nm) that self-assemble due to the hydrophobic effect. Both oil-in-water (O/W) and water-in-oil (W/O) microemulsions can be created, but the latter are most suitable for encapsulating hydrophilic substances like most BPPs [87,88]. W/O microemulsions consist of small surfactant-coated water droplets that are dispersed within an oil phase (Figure 1). The hydrophilic BPPs can be dissolved within the water droplets. Previous research has demonstrated that chymotrypsin and lysozyme can both be trapped inside W/O microemulsions and still retain their enzymatic activity [89]. Some of the advantages of microemulsions are that they are thermodynamically stable systems that can often be prepared using straightforward processing operations, such as simply mixing of the different components. The main limitation of microemulsions is that they typically require high levels of synthetic surfactants to formulate them. Another limitation of W/O-type microemulsions is that they can only be used in oral formulations that consist predominantly of oil, such as oil-filled soft capsules. Nevertheless, water-dispersible forms can be obtained by homogenizing the W/O microemulsion with water and a hydrophilic emulsifier to form a W/O/W type system [90,91] (Figure 1). Researchers have successfully encapsulated both BSA and cytochrome C in this kind of emulsified microemulsion, thereby protecting them from any stressors in the external aqueous phase [92]. It is possible to encapsulate some BPPs within the oil droplets in O/W microemulsions if they are naturally hydrophobic or they can be made hydrophobic, e.g., by hydrophobic ion pairing (HIP) [93]. Typically, HIP involves forming a complex between the charged BPPs and oppositely charged surfactants. In the pharmaceutical industry, one of the most successful means of orally delivering BPPs has been using self-emulsifying drug delivery systems (SEDDS) [93,94,95]. In these systems, BPPs are typically mixed with a hydrophobic surfactant and possibly a co-surfactant and/or lipid. When this mixture encounters aqueous gastrointestinal fluids, either a microemulsion or nanoemulsion is spontaneously formed, which encapsulates and protects the BPPs within the GIT, thereby leading to enhanced absorption [93]. It is important to design the system so that the BPPs remain trapped inside the oil droplets, otherwise they will not be protected [94]. Commercial products have been developed based on this technology, such as Neoral^®^ from Novartis (Cambridge, MA, USA), which is used to encapsulate cyclosporine (a bioactive cyclic polypeptide).

### 6.2. Emulsions

Typically, emulsions are formed by blending oil and water together to form a fine colloidal suspension of one of the liquids in the other [96]. Emulsions are thermodynamically unstable systems because the contact of oil and water is unfavorable due to the hydrophobic effect. Both O/W and W/O emulsions can be prepared, but the latter type is most suitable for encapsulating hydrophilic BPPs [97]. Nevertheless, W/O emulsions are only suitable for encapsulating BPPs in oral delivery systems that are primarily oil, such as oil-filled capsules [96]. BPPs can be incorporated into O/W emulsions if they are first converted into a hydrophobic form using the HIP approach discussed in the previous section. The BPPs can then be dispersed into an oil phase that is homogenized with an aqueous phase containing a hydrophilic emulsifier to create an O/W emulsion. In addition, W/O/W emulsions that can be dispersed in an aqueous environment can be formed by homogenizing W/O emulsions with water and a hydrophilic emulsifier [98]. In this case, the BPPs are encapsulated inside the small water droplets inside the larger oil droplets (Figure 1). W/O/W emulsions have been successfully employed to encapsulate insulin [99,100,101]. This kind of system should be able to protect this bioactive peptide from degradation in the mouth and stomach, but then release it in the small intestine where it can be absorbed. The main disadvantages of utilizing W/O/W emulsions for this purpose are that they are more costly and laborious to manufacture, requiring two homogenization steps and two emulsifiers (one hydrophilic and one hydrophobic), and they are often highly susceptible to breakdown during storage or when exposed to environmental stresses [102,103]. On the other hand, they can be formulated from a wide range of natural emulsifiers and oils, which is an advantage over most microemulsion systems.

### 6.3. Solid Lipid Particles

Traditionally, this kind of colloidal delivery system consists of fully or partially crystallized lipid particles dispersed in an aqueous medium [104,105]. As described earlier, BPPs can be complexed with oppositely charged surfactants using the HIP approach to make them hydrophobic, and then dispersed within a molten oil phase. The molten oil phase, a hydrophilic emulsifier, and water can then be homogenized to form an O/W emulsion or nanoemulsion, which is then cooled below the crystallization temperature of the lipid phase to promote the formation of solid lipid particles or nanoparticles. For these systems, it is important to select a lipid phase that is solid at ambient temperature but liquid at the emulsion preparation temperature. Alternatively, structured solid lipid particles suitable for encapsulating hydrophilic BPPs can be fabricated by preparing a W/O/W emulsion and then crystallizing the lipid phase [104,105]. Again, a lipid phase is used that is solid within the final W/O/W, but liquid during the emulsion preparation process [106,107]. This type of W/O/W emulsion can be formed using lipids that can be melted or dissolved in an organic solvent during homogenization. The lipids are then converted to solids by cooling or removing the organic solvent, although a requirement is that the BPP cargo must be able to withstand such temperature or solvent exposure. As a result, a solidified lipid phase is formed around the BPP-loaded water droplets (Figure 1). Solid lipid particles formed from W/O/W emulsion templates have been used to encapsulate insulin [106,108,109], thereby enhancing its oral bioavailability [106]. As with conventional W/O/W emulsions, a major disadvantage of this type of delivery system is that it is costly and laborious to produce, and it is susceptible to breakdown during storage.

### 6.4. Liposomes

Liposomes are comprised of one or more phospholipid bilayers held together by hydrophobic interactions between the non-polar tails (Figure 1) [110,111,112]. They are able to trap hydrophilic BPPs within the water core or between the polar head groups in the phospholipid bilayers [113]. BPPs with hydrophobic regions, including those naturally located in cell walls, can also be trapped in the phospholipid bilayers. A potential advantage of using liposomes is that they are comprised of phospholipids, which may lead to enhanced cell membrane permeability [114]. An important limitation of liposomes as delivery systems is that they typically have a low encapsulation efficiency (a large fraction of the BPPs is not trapped inside the aqueous interior) and are quite fragile (they easily break down during storage or before they reach the small intestine) [113,115,116]. Liposomes have been widely employed to encapsulate, protect, and delivery BPPs in pharmaceutical applications [1,117,118]. For instance, an appetite-stimulating hormone (ghrelin) has been encapsulated in liposomes, which enhanced its biological activity [119]. BPPs derived from hydrolysis of whey protein have also been encapsulated in liposomes [120]. A rat study, showed that encapsulation of insulin within liposomes enhanced its ability to lower blood glucose levels after oral administration [121].

### 6.5. Biopolymer Microgels

Biopolymer microgels are small particles containing a network of cross-linked polymer molecules, which are often proteins, polysaccharides, or their combination in food applications [122,123] (Figure 1). BPPs are trapped inside microgels by blending them with the biopolymers before microgel formation, or by loading them after microgel formation. Particularly for microgel formulations where the BPPs are loaded in after formation, there is the potential to achieve higher loading efficiencies (LE) due to favorable partitioning, as compared with emulsion or liposomal formulations. This type of colloidal particle can be constructed using a number of fabrication approaches, such as antisolvent precipitation, coacervation, emulsion templating, and injection-gelation [27,122,124]. The functional performance of biopolymer microgels can be tailored for particular applications by carefully controlling their compositions, dimensions, morphologies, porosities, and interfacial attributes. A potential limitation of microgels is that they are often quite porous, so that BPPs (particularly small peptides) can easily diffuse out of them. This problem can be overcome by ensuring that the pores are small enough to trap the BPPs or by having an attractive interaction between the BPPs and the biopolymer network inside the microgels [124]. The dependence of the BPP-biopolymer interactions on environmental conditions, such as pH, ionic strength, or temperature, can be used to form triggered delivery systems, which may be an advantage for some applications. Enzyme-triggered systems can also be produced by constructing the microgels from different kinds of biopolymers: starch (amylase), proteins (proteases), or dietary fibers (colonic enzymes).

Biopolymer microgels constructed from anionic polysaccharides have been employed to encapsulate both lipase and lactase, thereby enhancing their stability [78,79,125]. For instance, the gastric stability of the enzymes was increased by trapping them inside biopolymer microgels with an antacid [126]. BSA has been trapped inside alginate microgels by coating them with successive layers of cationic (chitosan) and anionic (dextran sulfate) biopolymers [127]. Insulin has been trapped inside hydroxypropyl cellulose-polyglutamic acid microgels, which enhanced its stability under gastric conditions [128].

## 7. Applications

There have been a large number of academic and commercial studies of the potential application of CDSs for encapsulation, protection, and delivery of peptides and proteins. In this section, only a few examples are given to highlight the potential of different kinds of CDSs for this purpose. Most of the information on this topic comes from pre-clinical studies, but a number of CDSs have been developed that are in various stages of clinical testing or are available commercially, which have been reviewed by other authors [129,130]. For instance, these authors reported that liposomes have been used to encapsulate insulin, anticancer peptides, and virus antigens, but not all of these applications were for oral delivery. In the future, however, it may be possible to use some of the same technologies to create oral formulations. Moreover, microemulsions have been developed to encapsulate insulin (Phase I trials), while SEDDs have been used to encapsulate cyclosporine to prevent organ rejection after organ transplants (commercially available as Neoral^®^).

### 7.1. Hormones

In this section, the encapsulation, protection, and delivery of insulin is used as an example of the utilization of CDSs for hormones. Insulin is administered to patients that have diabetes so as to control the glucose levels in their blood. At present, this hormone is usually administered by injection, which can be painful and inconvenient to patients, thereby reducing compliance [131]. As a result, the medical industry is interested in developing insulin formulations that can be administered orally [132,133]. There are, however, many hurdles to the successful oral delivery of insulin associated with its tendency to be chemically or enzymatically degraded within the human gut, as well as its relatively poor absorption into the systemic circulation [4,134]. For this reason, many researchers are examining ways of overcoming this problem using CDSs. A well-designed CDS should ensure that the insulin remains stable throughout the lifetime of the commercial formulation; inhibit its breakdown in the upper gastrointestinal tract; release it inside the small intestine; and, promote its absorption through the gut lining.

A wide variety of different CDSs have been shown to be capable of creating insulin-formulations that can be administered orally [2,133,134,135,136]. In this section, only a few examples are given to highlight the potential of this approach. Insulin has been encapsulated in polysaccharide microgels fabricated using an emulsification-gelation method [137]. The microgels formed consisted of an insulin-loaded alginate core surrounded by a protective chitosan coating. The microgels inhibited the breakdown of insulin when exposed to simulated gastric fluids, but were able to release the insulin when exposed to simulated intestinal fluids. Oral administration of the insulin-loaded microgels to diabetic rats led to improved control of blood glucose levels. The emulsification-gelation method has also been used to create insulin-loaded microgels using alginate and dextran sulfate as construction materials [138]. Another fabrication method, known as ionic-gelation, has been used to create insulin-loaded microgels from chitosan and tripolyphosphate, which were then encapsulated within W/O microemulsions [139]. Oral administration of this delivery system to rats was again shown to lead to better control of blood glucose levels in the animals. Insulin-loaded microgels have also been formed from alginate and calcium using an emulsion-template method [140]. A simulated digestion study showed that the insulin remained trapped inside these particles under gastric conditions, but was released under small intestine conditions. Insulin has also been encapsulated inside tiny water droplets that are themselves trapped inside lipid particles made of solidified fat [106]. These CDSs were prepared using W/O/W emulsions as templates. Simulated digestion studies demonstrated that the encapsulated insulin was protected from degradation within gastric fluids, but release in small intestine fluids. Oral administration of the insulin-loaded particles to diabetic rats increased the oral bioavailability five-fold compared to free insulin. A number of other CDSs have also recently been shown to have potential for insulin delivery, including yeast-based microcapsules [141], protein-coated liposomes [142], self-emulsifying drug delivery systems (SEDDS) [143], and carbon nanoparticles [144].

### 7.2. Digestive Enzymes

Colloidal delivery systems may also be advantageous for people who are unable to naturally generate enough digestive enzymes in their gut (particularly the small intestine), and therefore suffer from malnutrition or gastrointestinal discomfort. As examples, lactase could be orally delivered to people suffering from lactose intolerance [145,146] whereas pancreatic lipase could be delivered to people suffering from pancreatitis [147]. The oral delivery of digestive enzymes is often challenging because there are chemically or physically altered as they pass through the human gut. In particular, they may be denatured when they encounter the enzyme-rich and highly acidic environment of the human stomach. These problems may be overcome by encapsulating the digestive enzymes in CDSs that inhibit their degradation within the stomach, but release them inside the small intestine. A few examples of attempts to develop CDSs for this purpose are highlighted in this section.

Powdered lactase has been encapsulated in small digestible fat droplets using emulsion technology [148]. The authors showed that the encapsulated lactase retained its activity when exposed to simulated gastrointestinal conditions. Lactase has also been encapsulated within hydrophobic polymer capsules produced using an emulsion-evaporation approach [149]. Using a simulated digestion model, the researchers showed that the capsules protected the lactase from degradation under stomach conditions, but released it under small intestine conditions. Lactase has also been encapsulated within biopolymer microgels assembled from calcium alginate [125]. These microgels also contained an insoluble antacid that slowly dissolves when it is exposed to acidic conditions. As a result, the pH inside the microgels remains relatively constant even when they are submerged within gastric fluids. This type of microgel has also been utilized to encapsulate and protect pancreatic lipase from degradation under gastric conditions [78]. Various other kinds of CDSs have also been developed to protect lactase under gastrointestinal conditions, including biopolymer microgels [125,150,151], lipid microparticles [152], and plant-based microcapsules [153].

### 7.3. Vaccines

As with insulin, it would be much more convenient and less painful to administer vaccines through the oral route rather than through intravenous injections. In order to generate an immune response, vaccines delivered orally should be absorbed by the M-cells lining the human gut [154]. As with other BPPs, the delivery of vaccines through the oral route is difficult because of their susceptibility to degradation under gastrointestinal conditions, and their relatively poor absorption [155]. For this reason, considerable efforts have been made to create vaccine-loaded CDSs that protect the vaccines in the human gut and target the M-cells [131,156,157].

Vaccine-loaded polymer microgels orally administered to mice to mice have been shown to have enhanced bioactivity compared to controls [158]. Similarly, vaccine-loaded liposomes have also been reported to enhance the vaccines bioactivity after oral administration to mice [157,159]. Vaccine-loaded microgels fabricated from a synthetic polymer were shown to be taken up by M-cells and simulate an immune response [154]. Various other kinds of CDSs have also been investigated for their potential of enhancing the efficacy of orally-administered vaccines, including starch microparticles [160], chitosan microparticles [161], hydrophobic polymer particles [162], phospholipid liposomes [157,163], W/O/W emulsions [164,165], and polymer nanoparticles [166].

### 7.4. Antimicrobials

Some BPPs exhibit strong antimicrobial activity and may therefore be used to prevent or control microbial infections, including lysozyme, nisin, and cell-penetrating peptides [167]. The effectiveness of these antimicrobial BPPs is often limited due to their susceptibility to chemical degradation or tendency to interact with other substances in their environment. For this reason, there has been interest in the development of CDSs that could help maintain or enhance their antimicrobial efficacy [168,169]. Antimicrobial BPPs encapsulated in W/O microemulsions have shown to have better activity against *E. coli* than free BPPs [170]. Encapsulation of nisin and lysozyme within phospholipid liposomes has been shown to improve their antimicrobial activity against *Listeria monocytogenes* [171]. Similarly, the antimicrobial activity of nisin against *Listeria monocytogenes* has been enhanced by encapsulating it within biopolymer nanoparticles [172]. Other kinds of CDSs have also been shown to be capable of improving the activity of specific antimicrobial proteins, such as nisin in liposomes [173], chitosan microcapsules [174,175], pectin microparticles [176], and alginate/pectin microparticles [177], as well as lysozyme in zein microparticles [178,179] and starch microgels [180].

### 7.5. ACE Inhibitors

Some BPPs are able to reduce blood pressure due to their Angiotensin I-Converting Enzyme (ACE) inhibitor activity [181,182,183]. These ACE-inhibitors may therefore be suitable for treating people who suffer from hypertension. The oral administration of these BPPs is a challenge due to their bitter taste, as well as the fact that they may be hydrolyzed within the human gut, thereby decreasing their bioactivity. Researchers are therefore working to develop CDSs that will mask their disagreeable taste, inhibit their degradation within the stomach, and then release them inside the small intestine where they can be absorbed. An alternative approach is to use delivery systems to control the hydrolysis of proteins within the human gut, so that ACE-inhibitors are actually produced at an appropriate location in the GIT.

ACE-inhibitors have been loaded into biopolymer microgels fabricated from chitosan and alginate using an emulsion-templating method [184]. Using a simulated digestion model, the researchers showed that the microgels protected the encapsulated BPPs from degradation in the stomach, but then released them in the small intestine. Researchers have also been examining the potential of other CDSs for encapsulating and delivering ACE-inhibitors, including chitosan nanoparticles [185] and liposomes [186].

## 8. Conclusions

This review article has discussed the potential application of colloidal delivery systems for the encapsulation, protection, and controlled release of BPPs, including hormones, digestive enzymes, vaccines, antimicrobials, and ACE inhibitors. Oral administration of many of these BPPs is currently difficult due to the fact that they may be denatured, hydrolyzed, or aggregated during passage through the human gut. Moreover, many of them may not be efficiently absorbed from the gut into the systemic circulation. In principle, a wide variety of different colloidal delivery systems may be utilized to overcome these challenges, such as microemulsions, emulsions, solid lipid particles, liposomes, and microgels. There is some supporting evidence from pre-clinical studies that these CDSs may be able to protect BPPs under gastrointestinal conditions and enhance their absorption, but there are only a limited number of commercial products that have actually successfully passed clinical trials. For instance, the cyclosporine-loaded SEDDS (Novartis’ Neoral) mentioned earlier are now commercially available, whereas an octreotide-loaded colloidal system consisting of hydrophilic polymer particles in an oily medium has just completed a Phase III trial (Chiasma) [187]. There are considerable differences in the efficacy, cost, ingredient composition, manufacturing requirements, stability, and quality attributes of these delivery systems. Currently, researchers are working to identify the most efficacious delivery systems for specific applications. There are, however, few studies where different kinds of colloidal delivery systems are compared with each other for specific applications. Clearly, further work is required in this area to establish the relative merits and limitations of each technology for specific proteins and peptides, and then to actually test them in clinical trials. If this research is successful, then well formulated colloidal delivery systems may enable bioactive proteins and peptides to be used in a wider range of applications than is currently possible. It should also be noted that a new generation of low molecular weight peptide drugs is currently being developed in the pharmaceutical industry (“macrocycles”), which may circumnavigate some of the problems associated with the delivery of traditional protein or peptide-based therapeutic [188]. Finally, it should be noted that most of the previous work has been done in the pharmaceutical area, but that the food industry may benefit from this knowledge when developing functional and medical foods that contain BPPs.

## Figures and Tables

**Figure 1 molecules-25-01161-f001:**
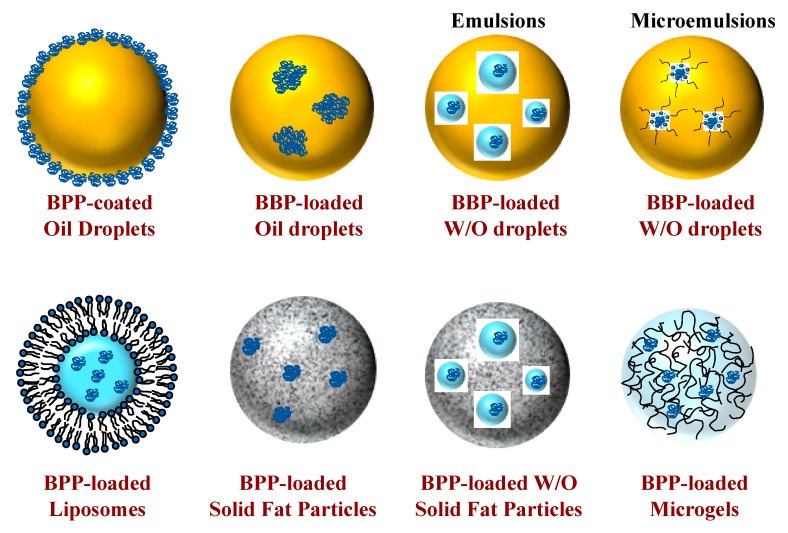
Schematic diagrams of colloidal delivery systems that could potentially be used to encapsulate, protect, and delivery hydrophilic bioactive proteins and peptides. All these systems could be dispersed in aqueous-based products.

**Figure 2 molecules-25-01161-f002:**
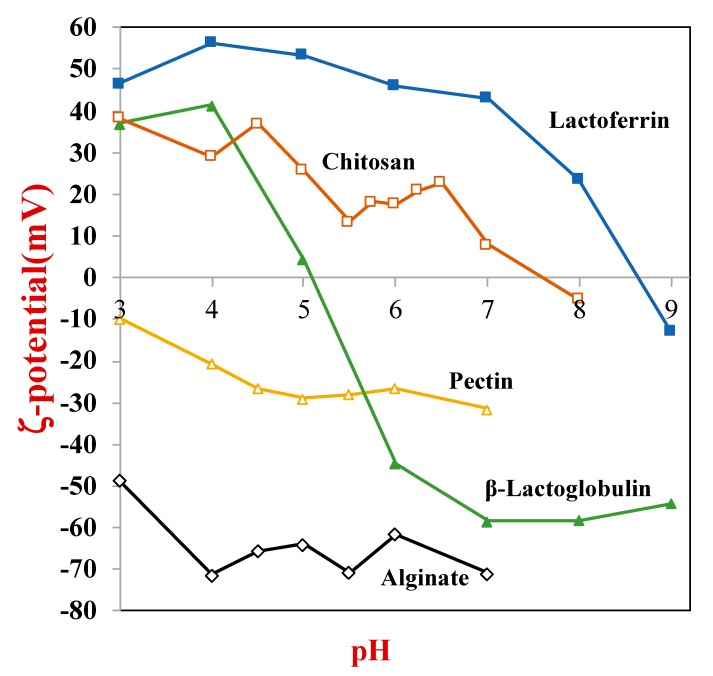
The electrical potential of biopolymers, such as proteins and polysaccharides, changes appreciably with pH due to ionization/deionization of charged groups.

**Figure 3 molecules-25-01161-f003:**
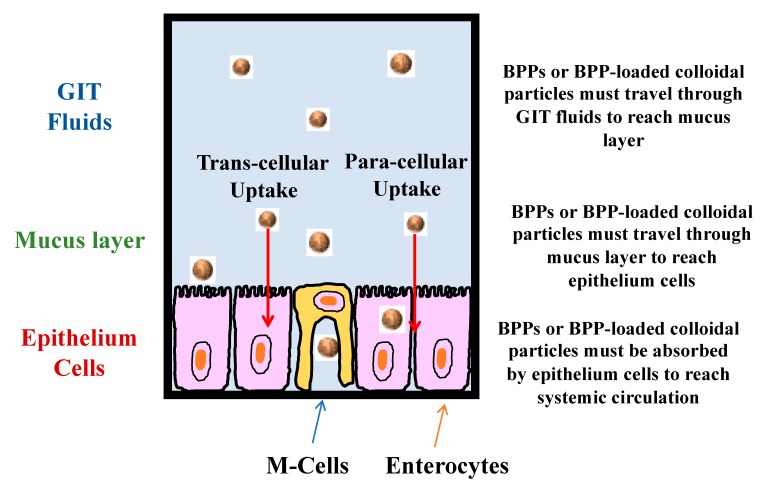
BBPs or BBP-loaded colloidal particles must move through the GIT fluids and mucus layer and then be absorbed by the epithelium cells before they can reach the systemic circulation. It should be noted that the M-cells are actually in the Peyer’s patches and only make up a small percentage (<5%) of the total intestinal cells. In practice, BPP-loaded colloidal particles are rarely absorbed through M-cells in vivo.

**Figure 4 molecules-25-01161-f004:**
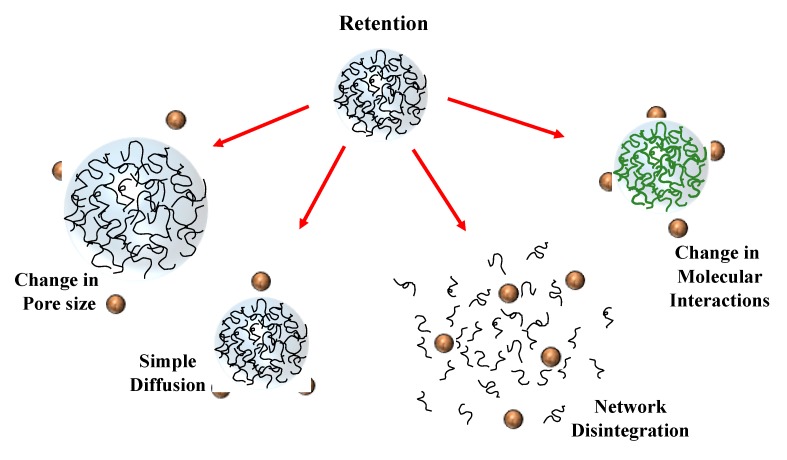
Colloidal particles can be designed to release encapsulated BPPs through various mechanisms, including changes in molecular interactions, pore size, network disintegration, or simple diffusion.

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
