# Peer review of "Recent Advances in Encapsulation, Protection, and Oral Delivery of Bioactive Proteins and Peptides using Colloidal Systems"

_molecules, 2020, doi:10.3390/molecules25051161_

Round 1
Reviewer 1 Report
The presented review paper is well organized and written in a clear language. It describes basic characteristics of different delivery systems for bioactive peptides and proteins, as abbreviated BPPs, and their possible applications. It will be very useful for non-specialists as it provides also general information on the important properties of a variety of colloidal delivery systems.
One major weakness of this manuscript is that it lacks a real discussion on the provided examples for delivering BPPs by different types of colloidal system and weighting their advantages and disadvantages (although the authors claim in the conclusion that such discussion is provided). In its present form, the manuscript gives rather general information and merely lists recent examples.
To improve the manuscript I would recommend the following:
1 - give some additional comments after each provided examples for application of delivery systems for BPPs in order to clearly outline its advantages and drawbacks (and try to avoid repetions)
2- rewrite the conclusion as to point out the basic tendencies for best/better performing colloidal system for specific cases
3- rewrite the abstract by shortening the general information and giving more details on the content and structure of the review paper
And some final comments on small mistakes: - BPPs appear also as BBPs (after p.10 in the manuscript and in Figs. 1 and 3, and in the abstract); in Fig. 1 there is identical naming for two different pictures (BPP-loaded solid fat particles) and this should be avoided.
After making the suggested corrections, I would agree that the manuscript is suitable for publication.
Author Response
Responses to Reviewer 1
- The authors thank the reviewer for their constructive comments and suggestions on our manuscript. We have revised the manuscript according to these comments and included a detailed list of responses below.
The presented review paper is well organized and written in a clear language. It describes basic characteristics of different delivery systems for bioactive peptides and proteins, as abbreviated BPPs, and their possible applications. It will be very useful for non-specialists as it provides also general information on the important properties of a variety of colloidal delivery systems.
- We thank the reviewer for their positive comments on our manuscript.
One major weakness of this manuscript is that it lacks a real discussion on the provided examples for delivering BPPs by different types of colloidal system and weighting their advantages and disadvantages (although the authors claim in the conclusion that such discussion is provided). In its present form, the manuscript gives rather general information and merely lists recent examples.
- As suggested, we have revised the manuscript to provide some more details about the relative merits and limitations of the different delivery systems covered.
To improve the manuscript, I would recommend the following:
1 - Give some additional comments after each provided examples for application of delivery systems for BPPs in order to clearly outline its advantages and drawbacks (and try to avoid repetitions)
- Please see previous comment.
2- rewrite the conclusion as to point out the basic tendencies for best/better performing colloidal system for specific cases
- As suggested, we have revised the conclusion to provide a more critical assessment of the different systems available. There have been very few studies where different kinds of colloidal delivery systems have been systematically compared for the same protein or peptide application. This is clearly an area where further research is needed, which has been pointed out in the revised manuscript.
3- Rewrite the abstract by shortening the general information and giving more details on the content and structure of the review paper
- As suggested, we have revised the abstract to provide more information about the structure of the manuscript.
And some final comments on small mistakes: - BPPs appear also as BBPs (after p.10 in the manuscript and in Figs. 1 and 3, and in the abstract); in Fig. 1 there is identical naming for two different pictures (BPP-loaded solid fat particles) and this should be avoided.
- As suggested, we have revised the manuscript to correct the BPPs and the figure caption.
After making the suggested corrections, I would agree that the manuscript is suitable for publication.
- Again, we thank the reviewer for their constructive comments on our manuscript.

Reviewer 2 Report
This is a very well structured review with clear messages. It is easy to read and provides the researchers with a brief overview of the progress in this area.
However, I have one critical comment/suggestion which should be considered, according to me, before the acceptance of this manuscript. I do not see reflected in the current manuscript the development of this hot area from the viewpoint of the real applications to humans.
There are several successful products on the market for oral delivery of digestive enzymes (lactases, combinations of enzymes) which are even not mentioned in the review. Clinical trials for oral delivery of insulin were also performed - are their lessons learned from these trials?
The existing positive (and negative) examples for application to humans deserve to be critically discussed, though briefly, to emphasize which are the real hurdles and how they have been overcome in the few successful cases known.
With its current content the manuscript is useful, but rather ordinary - one could find similar reviews in the literature. The authors, who are very competent and renowned scientists, may try to make a step further and complement the current text with a short analytical review on the application of these systems to human populations - in clynical studies and as commercial products. Such extension would provide significant added value to the current manuscript.
Author Response
Responses to Reviewer 2
- The authors thank the reviewer for their constructive comments and suggestions on our manuscript. We have revised the manuscript according to these comments and included a detailed list of responses below.
This is a very well structured review with clear messages. It is easy to read and provides the researchers with a brief overview of the progress in this area.
- We thank the reviewer for their positive comments on our manuscript.
However, I have one critical comment/suggestion which should be considered, according to me, before the acceptance of this manuscript. I do not see reflected in the current manuscript the development of this hot area from the viewpoint of the real applications to humans.
There are several successful products on the market for oral delivery of digestive enzymes (lactases, combinations of enzymes) which are even not mentioned in the review. Clinical trials for oral delivery of insulin were also performed - are their lessons learned from these trials?
- A main emphasis of the article was to develop food-based oral delivery systems that could be used for functional food applications (of which there is much less research on humans). We have therefore revised the manuscript to make this clearer (abstract and introduction). Nevertheless, we agree with the reviewer, that some valuable information can be obtained from previous studies on commercial pharmaceutical products. We have therefore revised the manuscript to include a brief discussion of these products and give references to other review articles that have discussed commercial applications.
The existing positive (and negative) examples for application to humans deserve to be critically discussed, though briefly, to emphasize which are the real hurdles and how they have been overcome in the few successful cases known.
- As suggested, we have included a brief discussion of the merits and limitations of the different delivery systems. The hurdles involved of very system specific, and we have highlighted some of the general hurdles in the article.
With its current content the manuscript is useful, but rather ordinary - one could find similar reviews in the literature. The authors, who are very competent and renowned scientists, may try to make a step further and complement the current text with a short analytical review on the application of these systems to human populations - in clinical studies and as commercial products. Such extension would provide significant added value to the current manuscript.
- We thank the reviewer for these suggestions. In the past, the main author has largely worked on the development of food-grade colloidal delivery systems suitable for application in functional and medical foods (rather than supplements or pharmaceuticals). I therefore have little experience in the area of commercial products in these areas. Instead, the purpose of the manuscript was to discuss delivery systems that could be assembled entirely from food grade ingredients. We have revised the manuscript to state this more clearly (abstract and introduction) and included a brief discussion of some commercial applications of colloidal delivery systems for proteins and peptides (application section).
Reviewer 3 Report
This review describes a range of colloid systems ranging from nanoparticles to microemulsions that can be used to deliver peptides and proteins by the oral route. It is very good on the detail of the colloids types and also the effects of pH and ionic strength on peptide charge in relation to pI. It is geared more for food scientists rather than Pharma, as there are few examples apart from insulin cited, but this is to be expected given the background of the authors. For example, the authors discuss bitter taste and also edible constructs, but this is not relevant for Pharma who make enteric coated dosage forms.Overall, there is a lot to like here, but it can be improved to have higher impact.
It would help in the Intro if the authors stated the different constraints of the Food and Pharma industries in relation to the biomaterials and excipients that are "allowed" by the FDA as against other bodies. The food industry is much more limited in this regard and the paper has very little on the Pharma side of oral peptide delivery where progress (and the science) actually far exceeds effort from the Food industry.
The MS could also do with more balance in relation to the permeability problem. At no point do they say how the colloids will enable this. The protection argument is well made, but most peptides will not cross unless enabled by permeation enhancers (See Rybelsus oral semaglutide from Novo-Nordisk). The argument for colloid uptake in vivo is very weak, so how will they achieve this? The authors agree that liposomes failed, so why will the other lipid-based systems work? Fig. 3 does not contribute much to this as it is theoretical and lacks in vivo support.
Also, an area missed is that local peptide delivery to the colon has been achieved for IBS - linaclotide, so there is potential for colloids to be used for local delivery for IBD etc, perhaps for antimicrobials for the colon. See recent review in ADDR by Bak, A. et al from Astra-Zeneca.
Some specific comments:
Line 15: a better phrase could be "BPPs presenting as ACE inhibitors....."
line 18: "storage or passage through the human gut" are separate concepts; this is confusing. Authors mean stability in storage.
Line 24: "...released in the intestine so it can be absorbed". This makes an assumption on permeability and also contradicts Fig 3 where they say the colloids are absorbed. Which is it? There is little evidence for either.
Line 62: BPP was already abbreviated at line 35
line 68: Don't understand the parameter referred to in relation to 3D structure
Line 114: Please define "polarity" for this wide readership
Line 158: Authors for the first time discuss permeability, but it is too brief and not balanced. There are some errors here. A "T-junction" is an odd phrase; authors mean tight junctions (TJs). They also over-emphasise M cells as a peptide route as they make up <5% of the FAE of PP, so the Fig 3 is misleading as M cells are mainly in PP not the regular epithelium. Oral vaccines targeting M cells has been around since the 90s, all efforts failed as particle uptake in humans is so low level. Also, efflux inhibitors refer to Pgp, not very relevant for peptides, except for hydrophobic cyclosporine.
line 171: in discussing product requirements, authors do not mention potency or half life - these are key considerations as to whether oral is commercially viable for a peptide.
line 243: The definitions of EE and LC are not the typical ones used for peptides entrapped in particles. I have never seen a definition of EE where the location in a colloid particle is defined this way. EE is the % peptide in the entire formulation, i.e. subtracting the free peptide in supernatant to indirectly calculate what is entrapped. LC is the ug peptide/mg particles, i.e. defined at the level of the particle. The authors have made this more convoluted that it needs to be.
-Conclusions: The authors assert that colloids can address the peptide absorption challenge, but there is no evidence presented to support this. They could cite cyclosporine SEDDS (Neoral, Novartis), or an oily emulsion of octreotide (Phase III, Chiasma), as examples of colloids in the clinic, but they have not. They also miss an opportunity to describe new types of stable LMW peptides being created in Pharma, macrocycles. These could go into these colloids with a better chance of success than trying to convert parenterals. The challenges for peptides will be less than for proteins give the MW difference, so the authors should be wary of lumping them together.
General comments:
-the legends to Figs are tiny.
-A Table of examples of peptides and proteins used in colloids in different research stages for different indications in different dosage forms would be helpful. This would also allow discrimination between food and Pharma peptide product types. The Delivery System Selection is quite light on examples. On SEDDS, there is a whole literature on octreotide and GLP-1 agonists.
Author Response
Responses to Reviewer 3
- The authors thank the reviewer for their constructive comments and suggestions on our manuscript. We have revised the manuscript according to these comments and included a detailed list of responses below.
This review describes a range of colloid systems ranging from nanoparticles to microemulsions that can be used to deliver peptides and proteins by the oral route. It is very good on the detail of the colloids types and also the effects of pH and ionic strength on peptide charge in relation to pI. It is geared more for food scientists rather than Pharma, as there are few examples apart from insulin cited, but this is to be expected given the background of the authors. For example, the authors discuss bitter taste and also edible constructs, but this is not relevant for Pharma who make enteric coated dosage forms. Overall, there is a lot to like here, but it can be improved to have higher impact.
It would help in the Intro if the authors stated the different constraints of the Food and Pharma industries in relation to the biomaterials and excipients that are "allowed" by the FDA as against other bodies. The food industry is much more limited in this regard and the paper has very little on the Pharma side of oral peptide delivery where progress (and the science) actually far exceeds effort from the Food industry.
- We thank the reviewer for this comment. Our research mainly focuses on the development of edible colloidal delivery systems for BPPs intended for oral ingestion via functional foods (rather than via drugs or supplements). As mentioned by the reviewer, the type of ingredients that can be used are much more limited in the food industry, which increases the challenges involved. In response to the reviewer’s suggestions, we have revised the manuscript to highlight this.
The MS could also do with more balance in relation to the permeability problem. At no point do they say how the colloids will enable this. The protection argument is well made, but most peptides will not cross unless enabled by permeation enhancers (See Rybelsus oral semaglutide from Novo-Nordisk). The argument for colloid uptake in vivo is very weak, so how will they achieve this? The authors agree that liposomes failed, so why will the other lipid-based systems work? Fig. 3 does not contribute much to this as it is theoretical and lacks in vivo support.
- We thank the reviewer for this valuable suggestion. In response, we have revised the manuscript to highlight the problems with permeability, and the need for permeation enhancers (which can be incorporated into CDS). We have also included the example of the commercial product mentioned.
Also, an area missed is that local peptide delivery to the colon has been achieved for IBS - linaclotide, so there is potential for colloids to be used for local delivery for IBD etc, perhaps for antimicrobials for the colon. See recent review in ADDR by Bak, A. et al from Astra-Zeneca.
- As suggested, we have included information about potential delivery of peptides and proteins to the colon in the revised manuscript.
Some specific comments:
Line 15: a better phrase could be "BPPs presenting as ACE inhibitors....."
- As suggested, we have revised this phrase.
line 18: "storage or passage through the human gut" are separate concepts; this is confusing. Authors mean stability in storage.
- As suggested, we have revised this phrase to clarify that we mean storage stability.
Line 24: "...released in the intestine so it can be absorbed". This makes an assumption on permeability and also contradicts Fig 3 where they say the colloids are absorbed. Which is it? There is little evidence for either.
- The absorption mechanism is likely to be highly system dependent. In some instances (e.g., indigestible colloidal particles), the whole particle may be absorbed. In other instances (e.g., digestible colloidal particles), the proteins may be released in the GIT fluids and then absorbed (but possibly hydrolyzed first). In response to an earlier comment from another reviewer we have removed the sentence mentioned from the abstract. We have also revised the manuscript to discuss this phenomenon in more detail (retention/release section).
Line 62: BPP was already abbreviated at line 35
- As suggested, we have removed this second definition of BPP.
line 68: Don't understand the parameter referred to in relation to 3D structure
- As suggested, we have clarified this statement (changing 3D structure to conformation)
Line 114: Please define "polarity" for this wide readership
- As suggested, we have included more information about the polarity of the proteins in the revised manuscript.
Line 158: Authors for the first time discuss permeability, but it is too brief and not balanced. There are some errors here. A "T-junction" is an odd phrase; authors mean tight junctions (TJs). They also over-emphasise M cells as a peptide route as they make up <5% of the FAE of PP, so the Fig 3 is misleading as M cells are mainly in PP not the regular epithelium. Oral vaccines targeting M cells has been around since the 90s, all efforts failed as particle uptake in humans is so low level. Also, efflux inhibitors refer to Pgp, not very relevant for peptides, except for hydrophobic cyclosporine.
- As suggested, we have revised this section to include more information about the permeability of the epithelium cells to proteins and peptides. We have also revised the discussion of TJs and M-cells.
line 171: in discussing product requirements, authors do not mention potency or half life - these are key considerations as to whether oral is commercially viable for a peptide.
- As suggested, we have included an additional section on the importance of “potency” and “half-life” in the revised manuscript.
line 243: The definitions of EE and LC are not the typical ones used for peptides entrapped in particles. I have never seen a definition of EE where the location in a colloid particle is defined this way. EE is the % peptide in the entire formulation, i.e. subtracting the free peptide in supernatant to indirectly calculate what is entrapped. LC is the ug peptide/mg particles, i.e. defined at the level of the particle. The authors have made this more convoluted that it needs to be.
- In response to the reviewers comments, we have revised the definitions for EE and LC.
-Conclusions: The authors assert that colloids can address the peptide absorption challenge, but there is no evidence presented to support this. They could cite cyclosporine SEDDS (Neoral, Novartis), or an oily emulsion of octreotide (Phase III, Chiasma), as examples of colloids in the clinic, but they have not. They also miss an opportunity to describe new types of stable LMW peptides being created in Pharma, macrocycles. These could go into these colloids with a better chance of success than trying to convert parenterals. The challenges for peptides will be less than for proteins give the MW difference, so the authors should be wary of lumping them together.
- We thank the reviewer for this comment. In response to the reviewer’s comments, we have revised the conclusions to mention these commercial products.
General comments:
-the legends to Figs are tiny.
- As suggested, we have increased the legend size
-A Table of examples of peptides and proteins used in colloids in different research stages for different indications in different dosage forms would be helpful. This would also allow discrimination between food and Pharma peptide product types. The Delivery System Selection is quite light on examples. On SEDDS, there is a whole literature on octreotide and GLP-1 agonists.
- As suggested, we have included an expanded discussion of SEDDS for delivery of peptides. As mentioned earlier, the main aim of this article was to introduce delivery systems that could be incorporated into functional foods, which have to be assembled from food-grade ingredients.
Round 2
Reviewer 1 Report
I accept the corrections made, and could recommend the manuscript for publication in its present form. It is an useful review paper.
Author Response
We thank the reviewer for their positive comments.
Reviewer 3 Report
The authors have addressed the vast majority of the comments made by the three Reviewers. It is much improved. In adding new material, there are a few very minor points that can be easily addressed.
Line 155: still do not understand the sentence on conformation and the sequence suggested. Sentence just needs to be better-written.
Line 227: Are mouth and stomach a target organ for BPPs? Just cite small intestine (for systemic) and colon (for local)
Line 273: Not aware that efflux inhibitors are used as a strategy to boost absorption of BPP (except for CsA), suggest omission.
Line 273: Reference for the Chiasma technology Phase III trial is: https://www.ncbi.nlm.nih.gov/pubmed/25664604. Ref for the technology: https://www.ncbi.nlm.nih.gov/pubmed/24558008
Line 286: source ref for oral semaglutide: https://www.ncbi.nlm.nih.gov/pubmed/30429357
Fig. 3 still irks and contradicts the text. In the image notes of Fig 3, maybe just re-state that uptake of BPP-loaded colloidal particles is likely to be a rare event in vivo. That would solve it.
line 615: Authors get in a bit of a knot over the pharmacological definitions of potency and half life. Potency is the concentration/dose that gives a half-maximal effect. The higher it is, the less peptide needed for the dosage form. Half-life refers to drug elimination half life in plasma once it gets there: influenced by BPP structure. Nothing to do with the colloid or residence time in the GIT.
Line 817: I do not believe that oral insulin ever reached Phase III trails in any format. Are the other two examples for oral? Mepact, and Inflexal. Not sure how relevant these are, but at least the route of delivery should be given.
In Conclusions, many examples are from Pharma, so it is appropriate to re-emphasise the applications for medical foods and functional foods, and that leveraging can occur.
Author Response
The authors have addressed the vast majority of the comments made by the three Reviewers. It is much improved. In adding new material, there are a few very minor points that can be easily addressed.
- Again, we thank the reviewer for their constructive criticism of our manuscript. We have revised the manuscript according to these comments and suggestions (please see responses below).
Line 155: still do not understand the sentence on conformation and the sequence suggested. Sentence just needs to be better-written.
- Our apologies, we have changed “conformation” to “three-dimensional structure”
Line 227: Are mouth and stomach a target organ for BPPs? Just cite small intestine (for systemic) and colon (for local)
- The mouth and stomach may be targets for some applications, for example antimicrobial peptides for oral health or certain peptides in functional foods to induce satiety and avoid overeating (for modulating hunger and satiety).
Line 273: Not aware that efflux inhibitors are used as a strategy to boost absorption of BPP (except for CsA), suggest omission.
- In the revised manuscript, we have highlighted that efflux inhibitors are not typically used to increase absorption of BPPs, with the exception of cyclosporine, which we have included as an example. We have also removed “efflux inhibitors” from the list of potential approaches.
Line 273: Reference for the Chiasma technology Phase III trial is: https://www.ncbi.nlm.nih.gov/pubmed/25664604. Ref for the technology: http://www.ncbi.nlm.nih.gov/pubmed/24558008
- We thank the reviewer for providing these references. We have included these references in the revised article.
Line 286: source ref for oral semaglutide: https://www.ncbi.nlm.nih.gov/pubmed/30429357
- We thank the reviewer for providing this reference, which has been included in the revised article.
Fig. 3 still irks and contradicts the text. In the image notes of Fig 3, maybe just re-state that uptake of BPP-loaded colloidal particles is likely to be a rare event in vivo. That would solve it.
- As suggested by the reviewer, this caveat has been included in the figure caption of the revised article.
line 615: Authors get in a bit of a knot over the pharmacological definitions of potency and half life. Potency is the concentration/dose that gives a half-maximal effect. The higher it is, the less peptide needed for the dosage form. Half-life refers to drug elimination half life in plasma once it gets there: influenced by BPP structure. Nothing to do with the colloid or residence time in the GIT.
- In response to the reviewer’s comment, the definitions of potency and half-life have been changed.
Line 817: I do not believe that oral insulin ever reached Phase III trails in any format. Are the other two examples for oral? Mepact, and Inflexal. Not sure how relevant these are, but at least the route of delivery should be given.
- As suggested, this sentence has been modified to remove drug names and indicate that many of the current formulations have been developed for non-oral administration. We have also stated that in the future, it may be possible to use some of the same technologies to create oral formulations.
In Conclusions, many examples are from Pharma, so it is appropriate to re-emphasise the applications for medical foods and functional foods, and that leveraging can occur.
- As suggested, we have revised the conclusions to highlight that most of previous work has been done in the pharma area, but that the food industry may benefit from this knowledge when developing functional and medical foods.